# Single-Sided Near-Field Wireless Power Transfer by A Three-Dimensional Coil Array

**DOI:** 10.3390/mi10030200

**Published:** 2019-03-21

**Authors:** Amirhossein Hajiaghajani, Seungyoung Ahn

**Affiliations:** 1Department of Electrical Engineering and Computer Science, University of California, Irvine, CA 92697, USA; ahajiagh@uci.edu; 2CCS Graduate School for Green Transportation, Korea Advanced Institute of Science and Technology (KAIST), Daejeon 305-701, Korea

**Keywords:** biomedical microrobot, magnetic devices, magnetic shielding, wireless power transfer

## Abstract

Wirelessly powered medical microrobots are often driven or localized by magnetic resonance imaging coils, whose signal-to-noise ratio is easily affected by the power transmitter coils that supply the microrobot. A controlled single-sided wireless power transmitter can enhance the imaging quality and suppress the radiation leakage. This paper presents a new form of electromagnet which automatically cancels the magnetic field to the back lobes by replacing the traditional circular coils with a three-dimensional (3D) coil scheme inspired by a generalized form of Halbach arrays. It is shown that, along with the miniaturization of the transmitter system, it allows for improved magnetic field intensity in the target side. Measurement of the produced magnetic patterns verifies that the power transfer to the back lobe is 15-fold smaller compared to the corresponding distance on the main lobe side, whilst maintaining a powering efficiency similar to that of conventional planar coils. To show the application of the proposed array, a wireless charging pad with an effective powering area of 144 cm^2^ is fabricated on 3D-assembled printed circuit boards. This 3D structure obviates the need for traditional magnetic shield materials that place limitations on the working frequency and suffer from non-linearity and hysteresis effects.

## 1. Introduction

Wireless power transfer (WPT) plays a key role in developing instruments that rely on external powering such as electric vehicles [1], micro-devices [2], and biomedical implants [3,4]. Over the past decade, new designs for coil structures have expanded the usage of WPT in various applications. Early wireless powering structures were based on the resonant coupling of aligned coils [5] and geometry optimization of the power transmitter (Tx) and receiver (Rx) coils [6]. Such structures generate intense magnetic fields over the surrounding space and are able to cause electromagnetic (EM) interference with nearby electric equipment [7,8]. In addition to non-thermal effects [9], an unintentional EM leakage can critically threaten several biomedical procedures such as electric pacemakers [10], magnetic drug targeting [11,12], and magnetic innervation [13]. In addition, several recent techniques for microrobot technology rely on WPT coils in combination with magnetic resonance imaging (MRI) coils for real-time feedback on the function and location of the microrobot [14,15,16,17,18]. However, the magnetic actuators responsible for wireless powering to microrobots often consist of pairs of Maxwell and Helmholtz coils that can easily affect the signal-to-noise ratio in the MRI’s radio-frequency pickup coils [19,20,21]. A single-sided WPT actuator that only affects the microrobot’s operation region and reduces the interference with the magnetic field of the MRI’s pickup coils can be a solution to this challenge (see Figure 1).

Moreover, in several applications such as wireless powering to biomedical devices [22,23], the Tx coils lie in the vicinity of lossy media such as living tissues. Conventional Tx coils create magnetic fields on either side of the transmitter, induce unintentional electric currents on the tissues, and have the potential to cause thermal damage [24]. Whilst magnetic sheets have been used to design shields to suppress the EM leakage in such conductive media [25], such techniques have limited the working frequency due to their non-linear behavior as well as complicated the cooling systems [26]. Generating a spatially controlled magnetic pattern for WPT that minimizes the risk of EM interference between multiple Rx coils has been a challenge for many biomedical systems.

To this aim, the spatial forming of subwavelength magnetic fields is a potential solution and may enable the control of the produced magnetic field and the formation of a single-sided radiation pattern aimed only at the Rx coils and away from the lossy media [12,27,28,29]. A traditional method to focus the magnetic field and suppress its leakage has been introduced by Halbach [30], which has since been the basis of a multitude of biomedical treatments such as magnetic drug delivery [31,32,33]. However, this scheme is based on arrays of permanent magnets and cannot be directly used for generating alternating magnetic fields. In a previous study, it was demonstrated that a combination of multi-turn coils in the form of linear Halbach arrays had the potential for lowering the EM leakage [34]. However, since bulky multi-turn coils generate enormous amounts of heat, such a structure cannot be used in compact designs such as charging pads and untethered powering to microrobots.

This paper is the first time a set of 3D coil arrays, based on the Halbach arrays and using vertical and horizontal circuit boards, has been demonstrated to improve the efficacy of low-leakage WPT using a single-sided radiation pattern. We aimed to minimize the magnetic field produced on the shielded side and reinforce it on the other side. The efficiency of the proposed design was validated through experimental tests at different distances from the Tx structure. The goal of this work was to investigate the practical potentiality of 3D coil arrays for low-leakage wireless powering. The total efficiency of the power transfer depends on the microrobot’s geometry, Rx coil design, and distance from the Tx. Compared to conventional wireless charging pads, the magnetic fields were blocked on one side of the introduced array of coils to lower the risk of EM interference.

## 2. Evolution of Halbach Arrays

Halbach arrays allow magnetic flow to be formed only on one side and weakens the magnetic field’s intensity on the opposing side. Generally, a coil-based scheme can substitute for a linear array of permanent magnets by considering the magnetization vector of each element and ensuring that the new coils generate a magnetic potential similar to the magnets. Based on the Halbach arrays, a one-dimensional (1D) coil scheme was introduced [34] (see Figure 2b), though it is ineligible for miniaturized wireless charging pads due to its bulky and difficult-to-fabricate structure. It also generates cross-polarized magnetic fields whose magnetic energy cannot be harvested even by a well-aligned Rx coil.

To reduce the cross-polarized fields and lower the overall weight of the Tx, the conventional linear Halbach chain was generalized to a planar two-dimensional (2D) array by adding perpendicular current paths. The schematic of the current in the unit element is shown in Figure 3, where the adjacent elements run currents in opposite directions. The magnetic near field of the path element running a current of *I* (phasor notation) is derived by [35]:(1)$$\vec{H_{c}}\left(x,y,z\right)=\frac{I}{4\pi }\int_{c}\frac{\vec{dl^{\prime }}\times \left(\vec{r}-\vec{r^{\prime }}\right)}{{\left|\vec{r}-\vec{r^{\prime }}\right|}^{3}}$$
where *r* and *r′* denote the vectors of the observation point and the point on the current path element *c* (shown in Figure 3).

The total magnetic field produced by curves with an array size of *M* × *N* (in *x* and *y* directions, respectively) is obtained by the superposition principle:(2)$${\vec{H}}_{total}=\sum_{m=0}^{M-1}\sum_{n=0}^{N-1}{\left(-1\right)}^{n+m}\vec{H_{c}}\left(x-nd,\;y-md,z\right).$$

The maximum side length of *d* depends on the working frequency such that the entire array’s size becomes significantly smaller than the wavelength. The minimum *d* is approximately equal to the distance between the Rx and Tx coils based on the analytical solution of magnetic fields generated by subwavelength coils [36]. The lateral loops are attached symmetrically (i.e., *a* = *b*, see Figure 3b).

The entire array scheme was modeled using the software COMSOL and the resultant spatial pattern of magnetic fields on a typical *y*-plane is shown in Figure 4b. It was observed that although no shielding material was employed, the main lobes of the magnetic fields take place only above the structure and play the role for power transfer. Meanwhile, the minor back lobes made by the lateral coils are unable to form an intense magnetic field or transfer power to nearby electric facilities under the Tx coils. Figure 4c compares the proposed single-sided magnetic pattern with the traditional planar 2D coils (i.e., the same structure without the lateral coils).

As shown in Figure 4b, the magnetic field’s intensity peaks at *z* = *a* while a near-zero field was observed at *z* = −*b*. Compared to traditional 2D actuators, the microrobot would receive a greater amplitude of magnetic fields above the structure (at *z* > *a* and running the same actuating current). By placing the actuator between the body and the pickup coils of the MRI, it was observed that the magnetic field’s intensity significantly decreases, which leads to minimizing the interference in the imaging coils. In addition, since the *x* and *y* components of the magnetic fields generated by the lateral coils cancel out each other in the main lobes, the cross polarization is insignificant.

## 3. Experimental Validation

In practice, the horizontal 2D structure was fabricated on a printed circuit board (PCB). Long narrow openings were incised in the horizontal PCB to form the vertical coils’ intersections. The vertical coils were printed on small circuit boards and slotted into the incisions. The connections were fixed and assembled by soldering the lateral and horizontal coils. The vertical coils were fixed by glue. All loops were serially connected and fed by a single power supply. The fabricated structure consisted of an array of 3 × 3 elements in *x* and *y* directions and its effective powering area was 144 cm^2^ (see Figure 5).

Since the proposed structure is designed to work through near-field coupling between the Rx/Tx coils, the maximum frequency is limited such that the overall length of the Tx coil remains much lower than the wavelength. However, there is no minimum frequency bound as the coils can generate single-sided DC magnetic fields. A small multi-turn Rx coil was used as the magnetic probe to measure the mutual inductance under and on top of the Tx. As the spatial pattern of the single-sided Tx system is non-uniform, the mutual inductance is dependent on the Rx location, however, we considered the peak values. Table 1 presents the measured inductance of the involved coils. 

Wirelessly powered microrobots often include a multi-turn receptor coil whose impedance is modified by a tuning capacitance. To verify the single-sided performance of the introduced Tx setup, the spatial magnetic pattern of the proposed design was measured by a magnetic probe with a design similar to that of a typical power receptor coil in a swimming microrobot. The probe was connected to a spectrum analyzer (GW INSTEK GSP-810, New Taipei City, Taiwan) as well as an oscilloscope (RIGOL Ds4012, Beijing, China) to measure the field’s intensity and polarity, respectively. A gridded polystyrene planar spacer was used to build different distance levels from the Tx. The Rx was positioned over the center of the gridded plane and manually scanned over the pad in 10 mm steps. The typical setup for a microrobot swimming above the Tx scheme and the magnetic field measurement setup are shown in Figure 6. 

The absolute efficiency of WPT in the strengthened side highly depends on the design of on the Rx structure and may not yield information about shielding. However, to evaluate the shielding function, the magnetic field’s intensity was measured at different distances from the coil array to calculate the relative WPT efficiency above and below the proposed 3D coil array.

According to the efficiency and shielding definition given by Paul [7], the WPT efficiency at different levels below the proposed Tx decreased 15-fold compared to the corresponding levels above the structure. With a shielding effectiveness of 11.7 dB for perpendicular magnetic fields, the proposed setup is a good solution for WPT to targets enclosed by radiation-sensitive electronic facilities. The measured radiation map (shown in Figure 7) had a dynamic range of 34 dB (from the pattern’s null to peak) and is in very good agreement with the simulation results.

The proposed structure is free of magnetizable elements such as ferrite shields and iron cores; hence, the actuating current can be maximized to the maximum tolerable temperature of the PCB substrate. Similar to other electromagnets, the current can be significantly increased by optimizing the trace width and thickness and by using cooling systems and particular substrates. As long as the overall length of the Tx coil is much lower than the wavelength of the working frequency, the effective powering area of this coil array can be extended in *x* and *y* directions. 

## 4. Conclusions

Planar coils printed on circuit boards have been the major building block of most of wireless power transfer systems for many years. In this study, a new type of power transmitter coil has been developed by employing 3D current routes instead of planar printed coils. The coil scheme includes out-of-plane elements and is assembled based on a generalized Halbach array, which allows for magnetic streams to construct and cancel on different sides of the actuator without the use of any shielding material. The controlled radiation pattern of the proposed structure will find a wide range of applications, from wireless powering to biomedical facilities utilizing MRI-driven microrobots in environments that are sensitive to electromagnetic interference. The proposed structure eliminates the need for traditional magnetic shields and enables the miniaturization of the wireless charging pads. The measurements confirm that the power level emitted through the back lobe was 11.7 dB lower than that of the front lobe. In addition to the quantified radiation pattern, the powering function of the 3D coil was verified by a microrobot swimming above the structure. This paper validates a general approach to the design of a low leakage power transmitter, though the powering efficiency can still be enhanced with regard to the particular needs of various applications. Other structures besides cubic arrays in a chessboard-shaped combination, such as an isometric or honeycomb structure, may also be employed.

## Figures and Tables

**Figure 1 micromachines-10-00200-f001:**
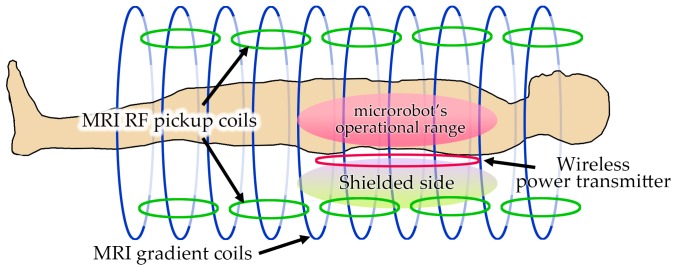
Schematic of the simultaneous usage of microrobot actuation and magnetic resonance imaging (MRI) coils to provide function and location feedback for implanted medical devices.

**Figure 2 micromachines-10-00200-f002:**
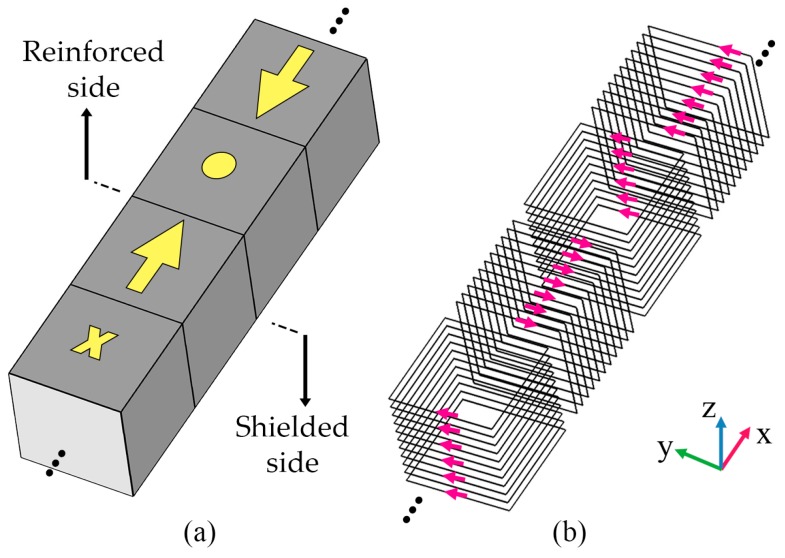
(**a**) A typical linear Halbach array in which the magnetization vectors are noted and (**b**) its coil-based equivalent.

**Figure 3 micromachines-10-00200-f003:**
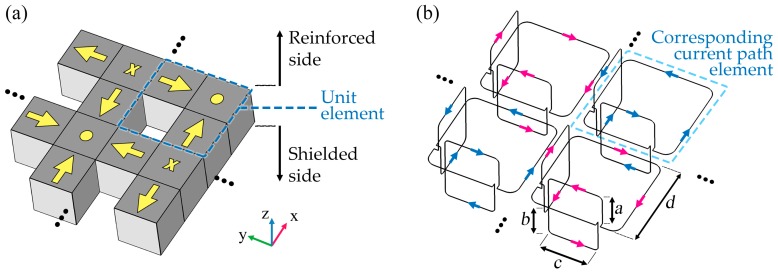
The geometry of the introduced 3D structure that represents an element of the planar array. The array may be extended in both *x* and *y* directions.

**Figure 4 micromachines-10-00200-f004:**
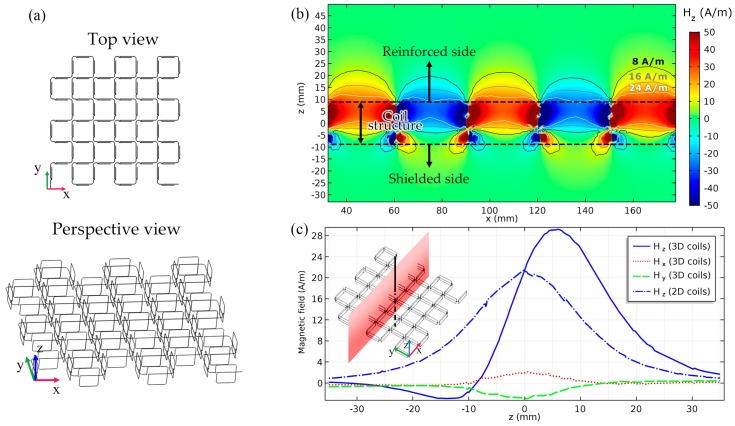
(**a**) Top and perspective views of the entire array scheme. (**b**) The intensity of the *z* component of magnetic fields in terms of phasors on the plane *y* = 10.5 cm for *a* = *b* = 6 mm, *d* = 30 mm, and *I* = 1 A. (**c**) A comparison between the proposed 3D coils and similar traditional 2D coils along the main lobe’s axis at *x* = *y* = 10.5 cm and *I* = 1 A. *z* < −7 mm and *z* > 7 mm indicate the shielded and strengthened zones, respectively. −7 mm < z < 7 mm is an inaccessible region where the structure is located.

**Figure 5 micromachines-10-00200-f005:**
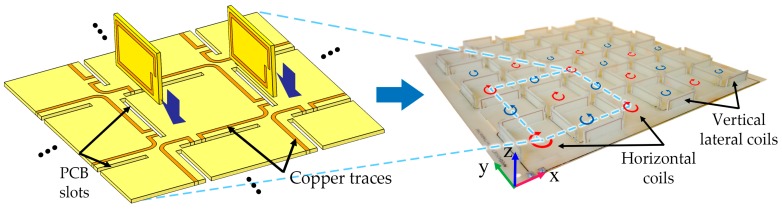
The vertical coils were slotted into the horizontal printed circuit board (PCB) openings. All the loops were serially connected. The connections were fixed by soldering and gluing.

**Figure 6 micromachines-10-00200-f006:**
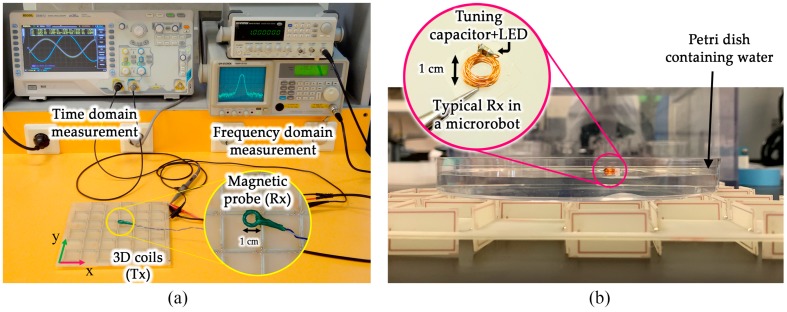
(**a**) The setup for measuring spatial patterns of the magnetic near-field at various distances from the transmitter (Tx) coil (gridded spacers not shown) at 1000 kHz. The magnetic probe had a similar structure to the microrobot’s power receiver coil. (**b**) The typical placement of a microrobot’s receiver (Rx) coil swimming on water. It is supplied by the single-sided power transmitter.

**Figure 7 micromachines-10-00200-f007:**
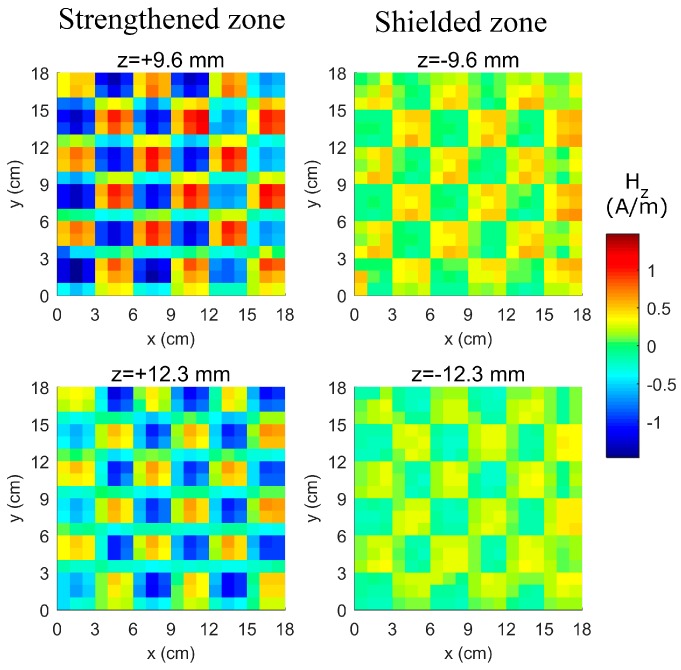
Magnetic radiation map at *z* = ±9.6 mm and ±12.3 mm. The power transferred to the shorted probe on the top side is 15 times greater than that transferred to the bottom side.

**Table 1 micromachines-10-00200-t001:** Measurement of the maximum inductance of the transmitter.

Object/Quantity	Rx Coil (Probe)	Tx Coil (3D Coils)
AC Resistance	0.5 Ω	7.8 Ω
Inductance	11 µH	9 µH
Mutual Ind. Peak at *z* = +9.6 mm	37.8 nH	
Mutual Ind. Peak at *z* = −9.6 mm	9.0 nH	

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
