# Peer review of "Single-Sided Near-Field Wireless Power Transfer by A Three-Dimensional Coil Array"

_micromachines, 2019, doi:10.3390/mi10030200_

Round 1
Reviewer 1 Report
The paper reports a 3D electromagnetic coil array based on the Halbach array to transmit power wirelessly with a single-sided radiation pattern. In general, the motivation of this work is clear, the 3D coil results are scientifically sound and interesting. However, some questions need to be clarify before the publish of the paper. My detailed comments are below:
1. In Fig. 4(c), an obvious difference in Hz is observed for 3D coils and 2D coils, the sign of the magnetic field changes. Will it rise some concerns to the application of wireless power transfer or for the control of microrobots? More discussions are needed for this figure.
2. Is the current passed through the 3D coil and the 2D coil the same and number of turns the same for comparison in Fig. 4(c)? Only with these conditions the comparison of absolute values are meaningful.
3. How is the PCB coils fabricated and assembled? More detailed methods need to be reported in the experimental section.
4. Are the vertical coils stable during the application of large current? What is the maximal field strength tested in the system? Will the coils deform or move due to the attractive or repulsive forces among each other? Is the design only suitable for high frequency application but not for DC applications?
5. How is the heat generation of the 3D coil compared to the 2D coil? Any cooling required?
6. English of the paper needs to be polished and many typos need to be checked carefully. For example, “cm2” on Line 22 Page 1, “such as living” on Line 48 Page 2, “very smaller than” Line 100 Page 4, and so on.
Author Response
The 'point-to-point reply' and 'the revised manuscript for review' are attached as a PDF document.

Reviewer 2 Report
In this study, a new wireless transfer approach is proposed. Aim was to prevent radiation leakage to certain sides of the system. To improve this manuscript further, following items are recommended.
Please discuss originality of the proposed idea explicitly in the introduction. Is there any similar works in literature? If there is anything comparable, can you compare your results with similar works in experimental results section?
This journal is about "micromachines" and the aim of the authors is to submit this work to special issue "implantable microdevices". By considering these two, it is expected from authors to perform their experiments on a micro-machine / micro-robot or a small scale mechanism to show suitability of the proposed system. Authors should demonstrate at least a preliminary experiments to convince reader that proposed system will be useful to transfer power to small smart systems safely.
Last but not least, it was good to see a clear and succinct explanation of work. Well written manuscript. Well done on that!
Author Response

(The authors gave the same response as above.)

Round 2
Reviewer 1 Report
The authors have made efforts to address the questions by the reviewer and the paper is improved and easier to follow by the readers. I would suggest acceptance of the paper.
Reviewer 2 Report
Questions are addressed well!